# Counterfactual Digital Twin: Generating What-If Trajectories with Uncertainty

## Abstract

Answering *what-if* questions is crucial in many decision-making domains, especially in time-sensitive areas such as healthcare, strategy, and policy. Generating counterfactual trajectories requires both predictions based on a unit's observed history and evidence from similar units exposed to the alternative conditions. Building upon this view, we develop an effective model called CounterTwin, which yields ensembles of all possible what-if trajectories and their posterior uncertainty. Specifically, CounterTwin learns how trajectories evolve from factual data using a transformer, and summarizes counterfactual evidence with synthetic control. These sources are then integrated through a Kalman filter, where the transformer serves as the prior belief and the synthetic control as a noisy measurement. This information fusion produces stable counterfactual rollouts with natural uncertainty. Extensive experiments on synthetic and real data show that CounterTwin achieves superior accuracy and robustness over existing methods. Code is available at https://anonymous.4open.science/r/CounterTwin-44F8/.

## 1 Introduction

Personalized decisions, such as choosing a medication or planning a care pathway, depend on anticipating how an individual will evolve over time. Digital twins promise exactly this prediction pathway by delivering individual-specific simulations that mirror the behavior of the real system (Batty, 2018; Mihai et al., 2022). We push that promise further by seeking answers to a counterfactual question "*What would have happened for the same individual under alternative conditions?*" In healthcare, clinicians wish to know how outcomes might change under a different treatment for a patient (Huang & Ning, 2012), and policymakers hope to evaluate the potential consequences of alternative policies (Jung et al., 1996). Yet answering these personalized what-if questions is intrinsically difficult because the counterfactual is never observed for the same unit, leaving no direct ground truth and inviting bias from confounding (Rubin, 1974; Pearl, 2009).

Many digital-twin systems learn replicas by fitting black-box generative models, such as generative adversarial networks (GANs), variational autoencoders (VAEs), or even large language models, directly to observed trajectories (Zhang et al., 2024; Jiang et al., 2021). While effective at imitation, these models optimize predictive fit rather than interventional semantics, and therefore fail to infer the counterparts of the replicas and show the link between actions and progression.

To infer counterfactual outcomes, one classical statistical approach is synthetic control (SC) (Abadie, 2021; Abadie et al., 2015; 2010; Billmeier & Nannicini, 2013), which constructs a counterfactual by identifying and aggregating trajectories from non-treated donor units that closely match the treated unit pre-intervention. Recent work couples SC with transformer architectures to estimate individual treatment effects. However, these methods often rely on strong structural assumptions (e.g., low-rank latent factors or linear mixing), require substantial regularization when donor coverage is limited, and are typically evaluated on continuous outcomes such as laboratory measurements (Bouttell et al., 2018; Qian et al., 2021).

We introduce *CounterTwin*, a counterfactual digital twin that unifies replica generation and counterfactual inference (Fig. 1). We train a quantile transformer on observed trajectories to learn individual dynamics and use it as a sequence prior at generation time. To drive counterfactual inference, we develop a method that learns donor information with mean and covariance based on SC, which do not rely on any assumptions on the feature structures. By serving as our evidence, it reveals how similar

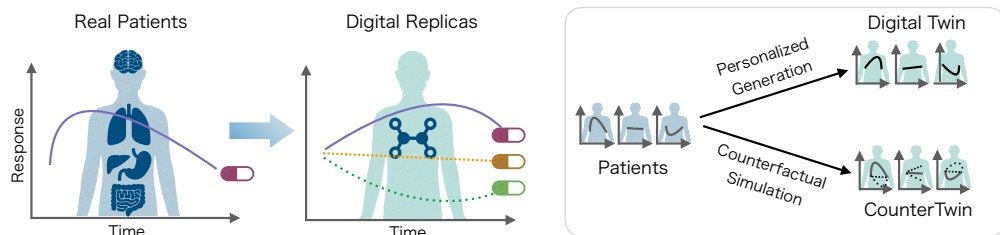

Figure 1: An illustration of the proposed CounterTwin, which generates personalized digital replicas (solid lines) for real agents (e.g., patients, areas) and what-if trajectories (dotted lines) to infer how the agent would evolve under alternative treatments.

units behave along alternative paths. We then fuse the predictive prior with donor evidence through a Kalman update. CounterTwin thus delivers factual replicas and credible what-if predictions within a Bayesian framework, which also results in posterior uncertainty in a natural way.

Our contributions are threefold: 1) We formalize a setting in which the digital twin must generate treatment-conditioned alternative paths (what-if replicas), rather than a single factual trajectory. 2) We present CounterTwin, which fuses quantile-level sequence predictions of individual progression with SC–based donor distributions, and integrates them via a Kalman update to produce multi-treatment replicas with posterior uncertainty. 3) Through extensive studies on synthetic simulations and real clinical data, CounterTwin yields trustworthy factual replicas and credible counterfactual trajectories, outperforming existing methods.

## 2 RELATED WORKS

**Synthetic electronic health record (EHR) generation.** Synthetic EHR generation enables data sharing while safeguarding privacy. Most methods learn generative models that approximate the joint distribution of longitudinal records. EVA (Biswal et al., 2021) uses a conditional VAE to synthesize patient trajectories. SynTeg (Zhang et al., 2021) adopts a two-stage pipeline in which a transformer captures the temporal structure and a conditional Wasserstein GAN produces visit sequences. PromptEHR (Wang & Sun, 2022) frames multimodal EHR synthesis as text-to-text generation, conditioning a BART backbone on patient attributes via prompts. KNNSampler (Beigi et al., 2022) constructs new patients by resampling features from nearest neighbors.

Digital twins (DTs) instantiate virtual patient replicas and generate new data, enabling real-time modeling of biological processes and in silico evaluation of candidate interventions (Laubenbacher et al., 2024; Kamel Boulos & Zhang, 2021; Zhang et al., 2024). For example, a recent method TWIN (Das et al., 2023) enables sample-efficient, personalized clinical trial DTs while preserving temporal and causal structures. However, most DT systems still produce a single trajectory per patient and do not support multi-treatment what-if simulations.

**Counterfactual prediction over time.** A complementary line of work constructs counterfactuals via synthetic control (SC). SyncTwin (Qian et al., 2021) learns time-invariant latent representations from pre-treatment windows and imputes post-intervention paths using donors matched in that space, while SCouT (Dedhia et al., 2022) maps pre-intervention spatio-temporal patterns to SC sequences with transformers. These approaches often rely on strong structural assumptions (e.g., low-rank latent factors or heavy regularization) and linear combinations of observed trajectories, which limits flexibility when donor coverage is sparse or treatment effects are nonlinear. Further, uncertainty is typically handled heuristically rather than via coherent predictive distributions.

Our CounterTwin differs from the aforementioned methods by fusing a learned sequence prior with donor-based evidence in a Bayesian manner, propagating both model and mismatch uncertainty, and yielding multiple treatment-conditioned counterfactual trajectories with calibrated predictive covariances. We summarize key differences of CounterTwin from related works in Table 1.

Table 1: Comparison of the proposed CounterTwin with related methods. SyncTwin (Qian et al., 2021) and SCouT (Dedhia et al., 2022) can only estimate continual what-if predictions ($\sqrt{}$).

| Methods | Reconstruction | Data Generation | What-if Prediction |
|---|---|---|---|
| EVA (2021) | $\sqrt{}$ | $\sqrt{}$ | $\times$ |
| PromptEHR (2022) | $\sqrt{}$ | $\sqrt{}$ | $\times$ |
| SynTEG (2021) | $\sqrt{}$ | $\times$ | $\sqrt{}$ |
| KNNSampler (2022) | $\sqrt{}$ | $\sqrt{}$ | $\times$ |
| TWIN (2023) | $\sqrt{}$ | $\sqrt{}$ | $\times$ |
| SyncTwin (2021) | $\times$ | $\times$ | $\sqrt{}$ (continuous only) |
| SCouT (2022) | $\times$ | $\times$ | $\sqrt{}$ (continuous only) |
| CounterTwin (Ours) | $\sqrt{}$ | $\sqrt{}$ | $\sqrt{}$ |

## 3 PROBLEM DEFINITION AND PRELIMINARIES

We study the digital twin generation problem under multi-valued, time-varying treatments. Let units $i \in \{1, \ldots, N\}$ (e.g., patients, regions) have static covariates $\mathbf{s}_i$ (e.g., sex, height). Over discrete time $t \in \mathcal{T}_i := \{0, \ldots, T_i\}$, unit $i$ receives treatment $d_{i,t} \in \mathcal{D}$, forming a treatment path $\mathbf{d}_{i,\mathcal{T}_i}$. Let $X_{i,0:T_0}$ be the input trajectory history and let $Y_{i,T_0:T_i}$ $(T_0 < T_i)$ denote the observed (factual) outcome trajectory under $\mathbf{d}_{i,\mathcal{T}}$. For any alternative path $\widetilde{\mathbf{d}}_{i,\mathcal{T}_i} \neq \mathbf{d}_{i,\mathcal{T}_i}$, we use $\widetilde{Y}_{i,T_0:T_i}(\widetilde{\mathbf{d}}_{i,\mathcal{T}_i})$ to represent the corresponding potential counterfactual trajectory.

Given the observed data $\mathcal{D}_{\mathrm{obs}} = \{(\mathbf{s}_i, \mathbf{x}_{i,0:T_0}, \mathbf{d}_{i,\mathcal{T}}, \mathbf{y}_{i,T_0:T_i})\}_{i=1}^{N}$, our goal is to learn a generative model $G_\theta$ that 1) produces unit-level 'what-if' twins $G_\theta(\widetilde{Y}_{i,T_0:T_i}(\hat{\mathbf{d}}_{i,\mathcal{T}_i})|\mathcal{D}_{\mathrm{obs}})$; 2) generate new, plausible units together with their potential trajectories as $G_\theta(\widetilde{Y}_{i,T_0:T_i}(\mathbf{d}_{i,\mathcal{T}_i}), \widetilde{\mathbf{s}}_i|\mathcal{D}_{\mathrm{obs}})$ to support downstream analyses. The fundamental challenge arises from the inherent limitation that, for each unit, only one (factual) trajectory can be observed. While the binary treatment case simplifies exposition, our framework addresses the more complex scenario involving multiple categorical treatments. Unlike typical digital twins, our task requires explicitly learning counterfactual trajectories under diverse treatment scenarios, which significantly increases the complexity.

### 3.1 SYNTHETIC CONTROL FOR COUNTERFACTUAL ESTIMATION

The synthetic control (SC) method (Abadie, 2021) constructs a counterfactual trajectory for a treated unit by forming a convex combination of untreated units. Let $y_{jt}^{(0)} \in \mathbb{R}$ denote the (untreated) outcome for unit $j \in \{1, \ldots, N\}$ at time $t \in \{1, \ldots, T\}$, with unit $j = 1$ exposed to treatment from $T_0 + 1$ onward and units $j = 2, \ldots, N$ remaining untreated. A standard linear factor representation is $y_{jt}^{(0)} = x_j^\top \beta_t + \mu_j^\top \lambda_t + \epsilon_{jt}$, where $x_j$ are observed predictors, $\mu_j$ are unobserved factors, $\beta_t, \lambda_t$ are time-varying loadings, and $\epsilon_{jt}$ is a mean-zero error.

SC chooses non-negative weights $\omega = [\omega_2, \ldots, \omega_N]^\top \in \mathbb{R}_+^{N-1}$ that sum to one and match pre-treatment predictors/outcomes as follows,

$$\sum_{j=2}^{N} \omega_j = 1, \qquad \sum_{j=2}^{N} \omega_j x_j = x_1, \qquad \sum_{j=2}^{N} \omega_j y_{jt} = y_{1t} \quad \text{for } t \leq T_0. \tag{1}$$

Given $\hat{\omega}$, the counterfactual (untreated) post-period path for the treated unit is predicted by the donor combination $\hat{y}_{1t}^{(0)} = \sum_{j=2}^{N} \hat{\omega}_j y_{jt}$, for $t > T_0$. Under the factor structure and sufficient pre-treatment fit, $\hat{y}_{1t}^{(0)}$ consistently approximates the untreated trajectory of the treated unit.

## 4 METHOD

We propose a novel framework, CounterTwin, designed to generate uncertainty-aware estimates of temporal counterfactual outcomes, for which the overall framework is shown in Fig. 2.

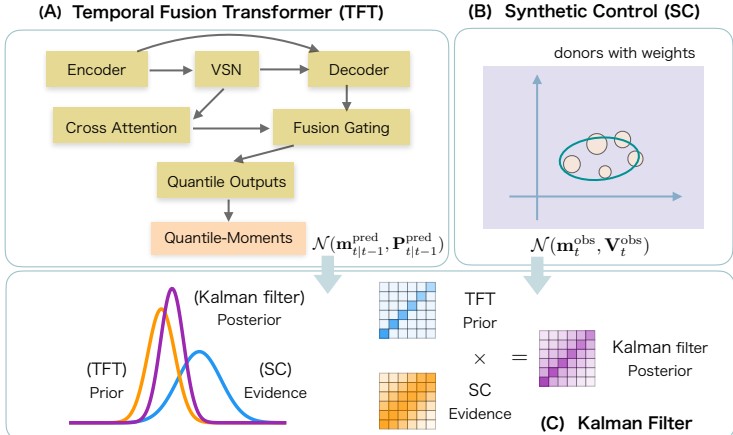

Figure 2: Overview of our CounterTwin framework. Panel A illustrates the TFT prior, which models the temporal evolution of patient states based on historical data. Panel B is our SC-based method, which collects information from similar samples. The Kalman filter in panel C fuses the TFT prior with SC evidence to produce the posterior counterfactual state (and uncertainty) at each step.

## 4.1 TEMPORAL FUSION TRANSFORMER (TFT) BASED PROBABILISTIC DYNAMIC LEARNER

We study units $i \in \{1, \ldots, N\}$ evolving over time $t$ under treatments $d_{i,t} \in \mathcal{D}$. We feed the TFT (Lim et al., 2020) the conditioning set $\mathcal{C}_{i,t} = \left(\mathbf{s}_i, \ \mathbf{X}_{i,0:T_0}, \ \mathbf{Y}_{i,T_0:t}, \ \mathbf{d}_{i,0:t}, \ \mathbf{d}_{i,t+1:T_i}\right)$, i.e., static covariates, observed histories, and a (factual or counterfactual) future treatment path. TFT builds a fused representation $\phi_{i,t+k} = f_\theta(\mathcal{C}_{i,t}, k)$ using a variable selection network (VSN), an encoder–decoder with attention, and gated residual connections, and outputs conditional quantiles at user-specified levels $\mathcal{Q} \subset (0,1)$ via parallel affine outputs $\widehat{y}_{i,t+k}^{(q)} = w_q^\top \phi_{i,t+k} + b_q$, where $q \in \mathcal{Q}$, $k = 1{:}H$, $w_q$ and $b_q$ also belongs to training parameters. All the parameters are trained end-to-end by the pinball loss

$$\mathcal{L}(\theta, \{w_q, b_q\}) = \sum_{i=1}^N \sum_{k=1}^H \sum_{q \in \mathcal{Q}} \rho_q\big(Y_{i,t+k} - \widehat{y}_{i,t+k}^{(q)}\big), \qquad \rho_q(u) = \max\{q\,u, (q-1)u\},$$

equivalently maximizing a quantile-regression (asymmetric–Laplace) likelihood. Hence $\{\widehat{y}_{i,t+k}^{(q)}\}$ are probabilistic predictions of $Q_Y(q \mid \mathcal{C}_{i,t}, k)$.

**From quantiles to moments (post–TFT).** To use the TFT outputs as prior, we convert quantiles to moments. Let $Q_Y(u)$ denote the conditional quantile function of $Y_{i,t+k} \mid \mathcal{C}_{i,t}$. The quantile function identities give

$$\mathbb{E}[Y] = \int_0^1 Q_Y(u)\,du, \qquad \mathrm{Var}(Y) = \int_0^1 [Q_Y(u)]^2\,du - \left(\int_0^1 Q_Y(u)\,du\right)^2.$$

Given TFT predictions at levels $0 < p_1 < \cdots < p_M < 1$ with values $y_j = \widehat{y}_{i,t+k}^{(p_j)}$, we define an extended grid with constant tails, $(p_0, y_0) = (0, y_1)$, $(p_{M+1}, y_{M+1}) = (1, y_M)$, and let $\widetilde{Q}_Y$ be the piecewise-linear interpolant on $\{(p_j, y_j)\}_{j=0}^{M+1}$. Applying the composite trapezoidal rule to $\widetilde{Q}_Y$ and $\widetilde{Q}_Y^2$ yields the estimates,

$$\widehat{\mu} = \widehat{\mathbb{E}}[Y] = \sum_{j=0}^M \frac{y_j + y_{j+1}}{2}\,(p_{j+1} - p_j), \ \widehat{\mathbb{E}}[Y^2] = \sum_{j=0}^M \frac{y_j^2 + y_{j+1}^2}{2}\,(p_{j+1} - p_j), \ \widehat{\sigma}^2 = \widehat{\mathbb{E}}[Y^2] - \widehat{\mu}^2.$$

The trapezoidal approximation is $O(h^2)$ with $h = \max_j(p_{j+1} - p_j)$, and sorting $\{p_j\}$ and reordering $\{y_j\}$ can improve numerical stability. These formulas provide per-unit, per-horizon $(\widehat{\mu}, \widehat{\sigma})$ directly from TFT's quantiles as

$$Y_{i,t+k} \mid \mathcal{C}_{i,t} \approx \mathcal{N}\big(\widehat{\mu}, \ \widehat{\sigma}^2\big). \tag{2}$$

## 4.2 SYNTHETIC CONTROL BEYOND LINEARITY

Classic SC rests on a linear factor view and returns a single point estimate (Eq. (1)). However, real-world dynamics are often nonlinear, and downstream fusion/decisions often require uncertainty. We therefore extend SC to (i) retain convex-hull matching in the pre-period while (ii) delivering post-period means and variances, including for categorical outputs represented in the logit space.

Let $y_{jt} \in \mathbb{R}^M$ stack $M$ outcomes (continuous and/or logit blocks). For a target unit and pre-period $t \leq T_0$, define $z_t \in \mathbb{R}^M$ (target) and $Z_t \in \mathbb{R}^{M \times J}$ (donors). We estimate convex weights $w \in \Delta^{J-1}$ via the scaled ridge least squares,

$$\hat{w} = \arg\min_{w \in \Delta} \sum_{t \leq T_0} \left\| S^{1/2}(z_t - Z_t w) \right\|_2^2 + \gamma \|w\|_2^2, \tag{3}$$

with $S = \mathrm{Diag}(s_1, \ldots, s_M)$ and $\gamma \geq 0$. Setting $S = I, \gamma = 0$ recovers the classic SC.

For $t > T_0$, we predict the counterfactual mean by

$$\widehat{\mu}_t = Z_t \hat{w}, \tag{4}$$

under the standard convex-hull condition and a sufficiently informative pre-period.

We explicitly propagate two uncertainty sources. Weight estimation captures sampling error from fitting the convex weights with a finite pre-period. Different pre-periods would yield slightly different $\hat{w}$. With the pre-period Gram matrix $\Phi = \sum_{t \leq T_0} Z_t^\top S Z_t$, we approximate the covariance of the fitted weights by

$$\widehat{\Sigma}_w \approx (\Phi + \gamma I)^{-1} \widehat{\sigma}^2, \qquad \widehat{\sigma}^2 = \frac{1}{MT_0} \sum_{t \leq T_0} \left\| S^{1/2}(z_t - Z_t \hat{w}) \right\|_2^2, \tag{5}$$

where $\widehat{\sigma}^2$ scales the overall noise level implied by the pre-period residuals after importance scaling.

Model mismatch captures the gap between the target's untreated path and the donors' convex span. From the pre-period residuals $r_t = z_t - Z_t \hat{w}$, we estimate the irreducible covariance,

$$\widehat{\Sigma}_{\mathrm{mis}} = \frac{1}{T_0 - 1} \sum_{t \leq T_0} (r_t - \bar{r})(r_t - \bar{r})^\top, \tag{6}$$

where $\bar{r}$ is the mean of $r$. It summarizes span mismatch and idiosyncratic variability that persists even with negligible weight error.

By linearizing $\widehat{\mu}_t = Z_t w$ around $w = \hat{w}$, the covariance yields

$$\widehat{\Sigma}_t = Z_t \widehat{\Sigma}_w Z_t^\top + \widehat{\Sigma}_{\mathrm{mis}}. \tag{7}$$

The first term propagates weight uncertainty through the post-period donor features; the second term adds the baseline misspecification identified in the pre-period. We summarize uncertainty with a convenient Gaussian working law, $y_t \sim \mathcal{N}(\widehat{\mu}_t, \widehat{\Sigma}_t)$.

For any categorical head with logits $\ell_t \in \mathbb{R}^K$, we apply Eqs. (4–7) on logits to obtain $(\mu_{\ell,t}, \Sigma_{\ell,t})$. Let $p_t = \mathrm{softmax}(\ell_t) \in \Delta^{K-1}$. A first-order multivariate delta method maps logit moments to probability moments,

$$\widehat{\mathbb{E}}[p_t] = \mathrm{softmax}(\mu_{\ell,t}), \qquad \widehat{\mathrm{Var}}(p_t) = J_t \Sigma_{\ell,t} J_t^\top,$$

where $J_t = \mathrm{Diag}(\hat{p}_t) - \hat{p}_t \hat{p}_t^\top$, the estimated probability is $p_t = \mathrm{softmax}(\mu_{\ell,t})$. This delivers calibrated class-probability means and covariances suitable for downstream fusion.

## 4.3 THE KALMAN FUSION OF TFT BELIEF AND SC EVIDENCE.

Up to this point, we have a predicted counterfactual trajectory as our state belief and synthetic-control outputs as noisy evidence. We apply the Kalman filter (Anderson & Moore, 2005) as a Bayesian fusion rule to combine them at each post-intervention time $T_0 < t < T_i$.

In the classic Kalman setup, dynamics follow $\mathbf{y}_t = \mathbf{F}_t \mathbf{y}_{t-1} + \boldsymbol{\varphi}_t$, where $\mathbf{F}_t$ is the state-transition matrix. In our case, the trained TFT (Eq. (2)) plays the role of the dynamics. We denote the

predictive mean and covariance as $\{\mathbf{m}^{\text{pred}}_{t|t-1}, \mathbf{P}^{\text{pred}}_{t|t-1}\}$ for the counterfactual state. The synthetic-control estimations (Eqs. (4–7)), denoted as $\{\mathbf{m}^{\text{obs}}_t, \mathbf{V}^{\text{obs}}_t\}$, act as measurements in the observation model $O_t = \mathbf{H}_t Y_t + \phi_t$. We take $\mathbf{H}_t = \mathbf{I}$ in our current experiments, while one can use a more complex matrix or even a network for more complex systems.

The Kalman update proceeds with the innovation and its covariance as follows,

$$\mathbf{r}_t = \mathbf{m}^{\text{obs}}_t - \mathbf{m}^{\text{pred}}_{t|t-1}, \qquad \mathbf{S}_t = \mathbf{V}^{\text{obs}}_t + \mathbf{P}^{\text{pred}}_{t|t-1},$$

and yields the gain and posterior,

$$\mathbf{G}_t = \mathbf{P}^{\text{pred}}_{t|t-1} \mathbf{S}^{-1}_t, \qquad \mathbf{m}_{t|t-1} = \mathbf{m}^{\text{pred}}_t + \mathbf{G}_t \mathbf{r}_t, \qquad \mathbf{P}_t = (\mathbf{I} - \mathbf{G}_t) \mathbf{P}^{\text{pred}}_{t|t-1}.$$

The uncertainty plays a key role here, as shown in panel (C) of Fig. 2. Smaller $\mathbf{V}^{\text{obs}}_t$ increases $\mathbf{G}_t$, pulling the estimate toward donor evidence, larger $\mathbf{V}^{\text{obs}}_t$ shrinks $\mathbf{G}_t$, keeping it near the model belief. The posterior covariance always contracts relative to $\mathbf{P}^{\text{pred}}_{t|t-1}$, so confidence grows only when evidence is informative. When the results of TFT and SC agree (i.e., $\mathbf{r}_t \approx \mathbf{0}$), the mean barely moves while uncertainty tightens; when they disagree, the shift scales with both the discrepancy $\mathbf{r}_t$ and the trust $\mathbf{G}_t$, preventing a noisy source from overruling a confident one. For categorical outputs (logit vectors), a full $\mathbf{V}^{\text{obs}}_t$ encodes cross-category couplings; the gain $\mathbf{G}_t$ redistributes evidence across coordinates so information about one category coherently adjusts its neighbors. Overall, the Kalman update acts as a principled arbiter between sources, automatically favoring whichever is more precise at each $t$ and yielding a natural uncertainty over the counterfactual trajectories.

## 5 EXPERIMENT

### 5.1 SIMULATION STUDY

**Fully synthetic data.** We simulate $N$ patients over $T$ visits, each assigned to a fixed treatment arm $c_i \in \{0, 1, 2\}$. Each patient has four static covariates scaled to $[0, 1]$, $\mathbf{x}_i = [\text{age}, \text{sex}, \text{bmi}, \text{com}] \in \mathbb{R}^4$, which both initialize and drive a 4D latent health state $\mathbf{s}_{i,t}$. A lab test value evolves via stable linear–Gaussian dynamics with treatment-specific drift and covariate effects, augmented by a nonlinear $\tanh$ curvature term and medication-specific persistence,

$$\mathbf{s}_{i,t+1} = A\mathbf{s}_{i,t} + B_{c_i} + C\mathbf{x}_i + \mathbf{u} \cdot \tanh(\mathbf{s}_{i,t}) + \text{effect}(\mathbf{s}_{i,t}) \cdot \mathbf{G}_{m_{i,t}} + \boldsymbol{\varepsilon}_{i,t},$$

where $\boldsymbol{\varepsilon}_{i,t} \sim \mathcal{N}(\mathbf{0}, \Sigma)$ and $\mathbf{s}_{i,0} \sim \mathcal{N}(C\mathbf{x}_i, 0.25^2 I)$.

Medications (Meds) are categorical $m_{i,t} \in \{0, 1, \ldots, K\}$ (with $0 = $ none), and their marginal impact is down-weighted by $\text{effect}(\mathbf{s}) \in [0, 1]$ (e.g., a logistic of a severity readout), yielding diminishing returns under severe disease. Adverse events (AEs) are categorical $a_{i,t} \in \{0, 1, \ldots, L\}$ with risk increasing in the level and worsening trend of $\mathbf{s}_{i,t}$ and shifted by medication. There are 8 types of AEs and Meds. Observed labs are a linear readout with an immediate medication offset and Gaussian noise: $y_{i,t} = \mathbf{h}^\top \mathbf{s}_{i,t} + w_{m_{i,t}} + \zeta_{i,t}$, with $\zeta_{i,t} \sim \mathcal{N}(0, \sigma^2)$. Together, this yields fully synthetic longitudinal EHRs where static features, treatments, medications, AEs, and labs are coherently coupled through a shared 4D process. (Details in Appendix.)

Specifically, we generate 480 samples and evenly input them into 3 treatment groups. Each of the time series has a total length of 100, where the first 10 visits are used as observed history and the rest are unobserved. We further divide each group into training and test sets with ratio $4 : 1$. For fully synthetic data, we generate all the three trajectories before-hand. Only the 'true' trajectories are used in training, while the other 'alternative' paths are used in evaluation.

**Evaluation.** We evaluate our CounterTwin on the following two tasks: 1) Reconstruction, which requires the model to imitate each unit, and 2) counterfactual generation, which requires the model to generate the alternative trajectories for each unit. We use the mean squared error (MSE) to evaluate the generated results on the lab test trajectory, and use the Hamming distance (HD) for the binary values, which is defined as the average number of mismatched events per visit.

On the reconstruction task, CounterTwin outperforms all baselines except for KNNSampler (Beigi et al., 2022). As expected, KNNSampler attains the lowest error because it simply returns the nearest

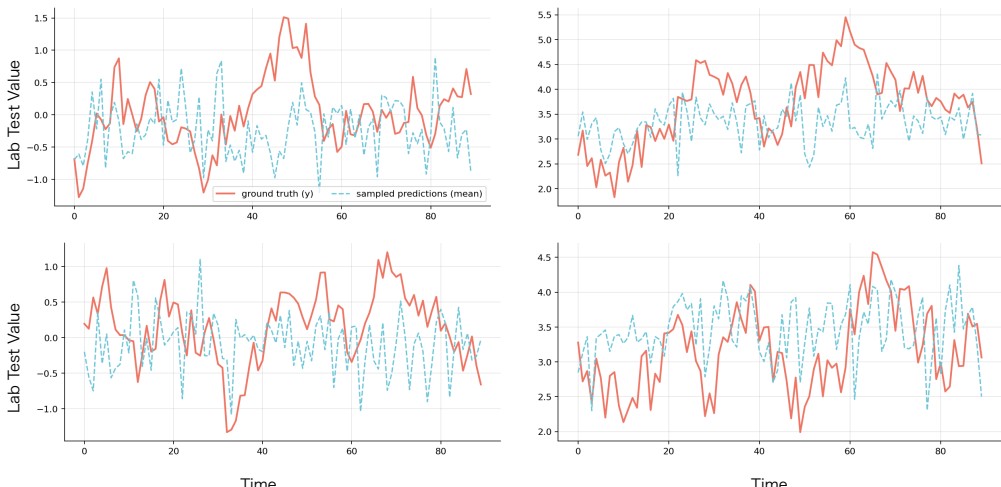

Figure 3: Counterfacutal predictions result examples on synthetic data. The predictions are generated via Monte Carlo sampling with 8 times.

Table 2: Evaluation results for patient data reconstruction and counterfactual trajectory generation on fully synthetic data. The metrics are Hamming distance (HD) for medication and adverse events, and MSE for lab test, with brackets showing standard errors. Lower values are better.

| Task | Model / Data | Medication | Adverse Event | Lab Test |
|---|---|---|---|---|
| *Reconstruction* | TWIN (2023) | $0.910_{(0.014)}$ | $0.207_{(0.011)}$ | $0.784_{(0.002)}$ |
| | KNNSampler (2022) | $\mathbf{0.216}_{(0.006)}$ | $\mathbf{0.161}_{(0.007)}$ | $\mathbf{0.158}_{(0.005)}$ |
| | EVA (2021) | $2.151_{(0.023)}$ | $1.571_{(0.012)}$ | $0.574_{(0.016)}$ |
| | PromptEHR (2022) | $0.830_{(0.014)}$ | $0.481_{(0.006)}$ | — |
| | CounterTwin (Ours) | $\mathbf{0.758}_{(0.060)}$ | $\mathbf{0.205}_{(0.008)}$ | $\mathbf{0.244}_{(0.004)}$ |
| *Counterfactual* | TFT (2020) | $0.947_{(0.004)}$ | $0.361_{(0.008)}$ | $0.816_{(0.006)}$ |
| | SynCtrl (2021) | $1.100_{(0.019)}$ | $0.483_{(0.012)}$ | $0.602_{(0.150)}$ |
| | CounterTwin (ours) | $\mathbf{0.839}_{(0.008)}$ | $\mathbf{0.303}_{(0.002)}$ | $\mathbf{0.601}_{(0.050)}$ |

visit as the 'reconstruction'. Nevertheless, our results show that CounterTwin produces high-fidelity replicas compared to other fully generative models.

On the counterfactual prediction task, we compare TFT, synthetic control with CounterTwin as an ablation study. Synthetic control performs better on continuous outcomes because it anchors each unit to a convex combination of well-matched donors, which is well aligned with the synthetic continuous lab test. By contrast, TFT excels at binary events. The attention and gating capture nonlinear, threshold-like interactions among covariates, treatment, and medication. Donor averaging in SC might wash out sharp onsets of rare events. Fusing both signals, CounterTwin treats TFT as a dynamic prior and SC as a low-variance observation in a Bayesian update, yielding the best overall performance. We show some of the counterfactual generations (compared with synthetic groundtruth) in Fig. 3.

## 5.2 EXPERIMENT ON REAL DATA

**Dataset description.** In a phase III breast cancer clinical trial (NCT00174655) involving 2,887 patients, participants were randomly assigned to four treatment groups: two active comparator groups where Doxorubicin with Cyclophosphamide was administered either sequentially or concurrently, and two experimental groups where Doxorubicin with Docetaxel was given either sequentially or concurrently. The primary objective was to compare disease-free survival (DFS) across these treatment strategies in the adjuvant treatment for patients with node-positive breast cancer.

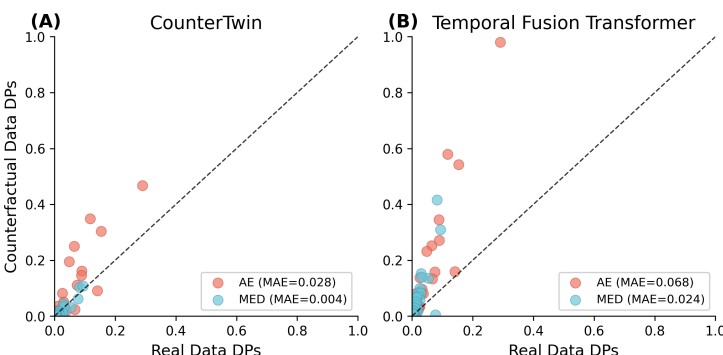

Figure 4: Dimension-wise probabilities (DPs) of medication and adverse events for the generated counterfactual data (control) vs. real data (treatment).

**Problem definition.** Given data $\mathcal{D}_{\text{obs}} = \{\mathbf{s}_i, \mathbf{d}_{i,\mathcal{T}_i}, \mathbf{Y}_{i,\mathcal{T}_i}\}_{i=1}^N$, where $N$ is the total sample size, $\mathbf{s}_i \in \mathbb{R}^S$, $\mathbf{d}_{i,\mathcal{T}_i}$, and $\mathbf{Y}_{i,\mathcal{T}_i} = (y_{i,1}, \ldots, y_{i,T_i})$ represent the static covariates, treatment trajectory, and longitudinal visit records for patient $i$, respectively. Each visit $y_{i,t}$ is a collection of $U$ types of events, $y_{n,t} = \{y_{n,t}^u\}_{u \in U}$. The event types include medication ($y_{n,t}^{\text{med}}$ or $y_{n,t}^1$), and adverse events ($y_{n,t}^{\text{AE}}$ or $y_{n,t}^2$). Each event type $y_{n,t}^u$ is represented as a multi-hot vector of dimension $l_u$, indicating the occurrence of specific events within that type at visit $t$.

**Reconstruction evaluation.** As the breast cancer trial dataset only involves binary events, we utilize Hamming distance (HD) to evaluate the reconstruction of patient visit sequences. Similar to the experiment on fully synthetic data, CounterTwin demonstrates superior performance over other generative baselines, except the KNN sampler (Table 3).

Table 3: Evaluation results for patient data reconstruction on the breast cancer trial dataset. The metric is Hamming distance (HD) for both medication and adverse events, with brackets showing standard errors. Lower values are better.

| Model / Data | Medication | Adverse Event |
|---|---|---|
| TWIN (2023) | $0.980_{(0.011)}$ | $1.876_{(0.011)}$ |
| KNNSampler (2022) | $\mathbf{0.383}_{(0.007)}$ | $\mathbf{1.291}_{(0.018)}$ |
| EVA (2021) | $6.847_{(0.070)}$ | $4.452_{(0.001)}$ |
| PromptEHR (2022) | $2.003_{(0.020)}$ | $3.283_{(0.066)}$ |
| CounterTwin (ours) | $\mathbf{0.895}_{(0.010)}$ | $\mathbf{1.368}_{(0.014)}$ |

**Counterfactual evaluation.** As the ground truth counterfactual outcomes are unavailable in the real dataset, patient-level evaluation of the generated counterfactuals is not feasible. We instead evaluate the quality of the counterfactuals by comparing the distributions of binary events within the generated data to those observed in the actual treatment cohort. Following Das et al. (2023), we employ dimension-wise probability (DP) to measure the prevalence of each individual event appearing in any given visit within the dataset. The DPs calculated from the generated counterfactual data are compared against the DPs calculated from real data. While TFT could preserve some correlation between generated and real event distributions, it tends to overestimate the DPs (Fig. 4). In comparison, CounterTwin shows more accurate matching of DPs with significantly lower mean absolute error (MAE), indicating alignment with real data.

## 6 A PRACTICAL APPLICATION ON COVID-19 DATA

We apply CounterTwin to COVID-19 case counts and policy indicators from the Oxford COVID-19 government response tracker (OxCGRT)[1] . Treating the issuance of *gathering restrictions* as the intervention date, we learn factual dynamics with a Temporal Fusion Transformer (TFT) prior and assemble donor evidence from comparable small nations/regions (e.g., Macao, Thailand, Taiwan).

---

[1]https://github.com/OxCGRT/covid-policy-tracker/tree/master

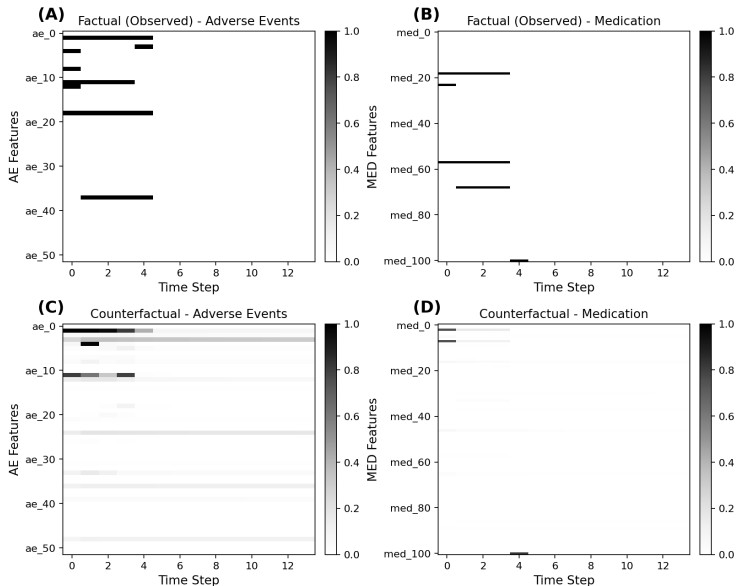

Figure 5: Factual (observed) and counterfactual event frequency heatmap for a random patient. Panels (A) and (B) display the observed prevalence of adverse events and medication. Panels (C) and (D) present the counterfactual predictions.

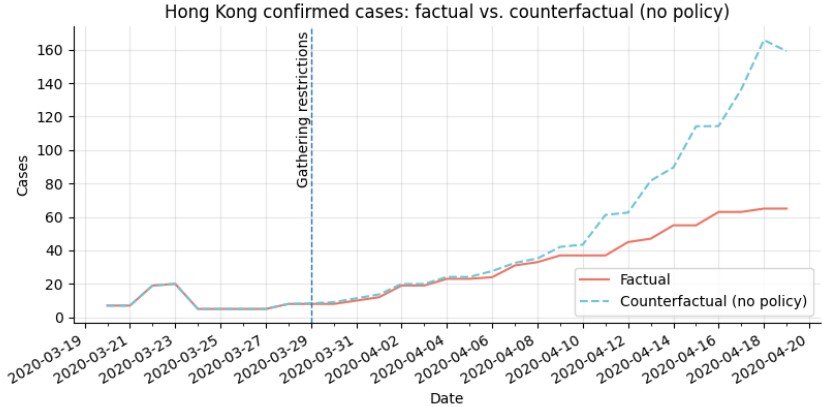

Figure 6: Apply the proposed CounterTwin on OxCGRT dataset.

We then fuse the TFT prior and donor signals via a Kalman-style Bayesian update to produce a counterfactual trajectory representing the "no-policy" scenario. The generated counterfactual trajectory shows how case numbers would have evolved had the restriction not been issued.

# 7 CONCLUSION

We address a key challenging problem in this paper, which require digital twins to generate counterfactual predictions, not merely imitate observed trajectories. We introduce CounterTwin, a practical framework that fuses a transformer prior with donor-based evidence via a Kalman-style Bayesian update. On fully synthetic and real clinical datasets, CounterTwin consistently matches or surpasses strong baselines, with ablation studies confirming the complementary strengths of the TFT prior and synthetic control. The method produces plausible, individualized counterfactual trajectories and scales to long horizons and multi-arm treatments.

ETHICS STATEMENT

We use public and fully synthetic dataset which do not have any re-identification risk, and will release code for synthetic data for reproducibility. CounterTwin is a research method, not a clinical decision system, and predictions are reported with uncertainty. Any practical use requires expert oversight, subgroup fairness checks, prospective validation, and regulatory review.

REPRODUCIBILITY STATEMENT

All experiments in CounterTwin are fully reproducible. We fix random seeds, log every hyperparameter, data split, and training configuration, with scripts to regenerate synthetic data and preprocess real datasets. We will release the complete codebase, including configuration files, experiment runners, and trained checkpoints upon publication.

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

## A  APPENDIX

### A.1  USE OF LLMS

LLMs were used solely for grammar and phrasing in this paper. No LLMs were used for ideation, code implementation, data analysis, or experiments.

### A.2  SYNTHETIC DATA GENERATION DETAILS

**Notation and dimensions**  Each patient $i$ is assigned a fixed treatment arm $c_i \in \mathcal{C} = \{0, 1, 2\}$, and we generate ground-truth outcomes for all three arms. At visit $t$, we produce:

- static covariates $\mathbf{x}_i \in \mathbb{R}^4$,
- latent state $\mathbf{s}_{i,c,t} \in \mathbb{R}^4$,
- laboratory measurement $y_{i,c,t} \in \mathbb{R}$,
- medication category $m_{i,c,t} \in \{0, 1, \ldots, K\}$ (0 = no medication, $K=10$),
- adverse-event (AE) category $a_{i,c,t} \in \{0, 1, \ldots, L\}$ (0 = no AE, $L=5$).

For brevity, we drop subscripts when unambiguous.

**Static covariates**  We draw age $\sim$ Unif$[20, 80]$, sex $\sim$ Bernoulli$(0.5)$, bmi $\sim \mathcal{N}(25, 4^2)$ clipped to $[15, 45]$, com $\sim$ Poisson$(1.5)$ capped at 5, and concatenate them into $\mathbf{x}_i$.

**Latent state dynamics** Let $A = \mathrm{diag}(0.92, 0.85, 0.80, 0.70)$, $\mathbf{u} = [-0.03,\ 0.02,\ 0,\ 0]^\top$, and $\Sigma = 0.04\,\mathbf{I}$. Medication has a persistent effect $\mathbf{G}_{m_t}$ modulated by disease severity through

$$\mathrm{eff}(\mathbf{s}_t) = \sigma\big(k_{\mathrm{eff}}(\tau - \mathrm{level}_t)\big), \qquad \mathrm{level}_t = \mathbf{d}^\top \mathbf{s}_t, \quad \mathbf{d} = [1,\ 0.3,\ 0.6,\ 0]^\top, \tag{8}$$

with logistic $\sigma(\cdot)$, threshold $\tau{=}2.0$, and slope $k_{\mathrm{eff}}{=}1.2$; thus $\mathrm{eff} \approx 1$ for mild disease and $\mathrm{eff} \to 0$ for severe disease. The state update is

$$\mathbf{s}_{t+1} = A\,\mathbf{s}_t + \mathbf{u} + \mathrm{eff}(\mathbf{s}_t)\,\mathbf{G}_{m_t} + \varepsilon_t, \qquad \varepsilon_t \sim \mathcal{N}(\mathbf{0}, \Sigma). \tag{9}$$

**Medication (categorical)** We first decide whether any medication is prescribed using the previous lab and current state:

$$\mathbb{P}(\mathrm{any\_med}_t{=}1) = \sigma\big(a_{\mathrm{pres}}^\top \mathbf{s}_t + \rho_{\mathrm{pres}}\, y_{t-1} + \eta_{\mathrm{pres}}\big), \tag{10}$$

where $a_{\mathrm{pres}} \in \mathbb{R}^4$ is small, $\rho_{\mathrm{pres}} > 0$ (higher prior lab $\Rightarrow$ more meds), and $\eta_{\mathrm{pres}}$ is a bias term. If $\mathrm{any\_med}_t{=}0$, then $m_t = 0$; otherwise we sample $m_t \in \{1, \ldots, K\}$ via

$$\mathbb{P}(m_t = k \mid \mathrm{any\_med}_t{=}1) = \frac{\exp\{(W_{\mathrm{med}}\mathbf{s}_t + \mathbf{b}_{\mathrm{med}})_k\}}{\sum_{j=1}^{K} \exp\{(W_{\mathrm{med}}\mathbf{s}_t + \mathbf{b}_{\mathrm{med}})_j\}}, \tag{11}$$

with $W_{\mathrm{med}} \in \mathbb{R}^{K \times 4}$ and $\mathbf{b}_{\mathrm{med}} \in \mathbb{R}^K$. Each category $k$ has a persistent vector $\mathbf{G}_k$ and an immediate lab offset $w_k$ (with $\mathbf{G}_0 = \mathbf{0}$ and $w_0 = 0$).

**Adverse events** We model AEs with an any-AE Bernoulli followed by a type multinomial when an AE occurs. Define the short-term trend

$$\mathrm{trend}_t = \begin{cases} \mathbf{d}^\top (\mathbf{s}_t - \mathbf{s}_{t-1}), & t > 0, \\ 0, & t = 0\ , \end{cases}$$

and let worse level/trend increase risk while treatment and medication shift log-odds:

$$\mathbb{P}(\mathrm{any\_AE}_t{=}1) = \sigma\big(a_1\, \mathrm{level}_t + a_2\, \mathrm{trend}_t + \kappa_c^{\mathrm{any}} + \xi^{\mathrm{any}} + \Delta_{m_t}^{\mathrm{any}}\big), \tag{12}$$

with $a_1, a_2 > 0$, arm-specific offset $\kappa_c^{\mathrm{any}}$ (beneficial arm 1: $\kappa_1^{\mathrm{any}} < 0$; adverse arm 2: $\kappa_2^{\mathrm{any}} > 0$), baseline $\xi^{\mathrm{any}}$, and medication suppression $\Delta_m^{\mathrm{any}} \leq 0$ (with $\Delta_0^{\mathrm{any}} = 0$). Conditional on $\mathrm{any\_AE}_t{=}1$, we draw $a_t \in \{1, \ldots, L\}$ via

$$\mathbb{P}(a_t = \ell \mid \mathrm{any\_AE}_t{=}1) = \frac{\exp\{(U_{\mathrm{AE}}\mathbf{s}_t)_\ell + \beta_{\mathrm{trend}}^{(\ell)}\mathrm{trend}_t + \kappa_{c,\ell}^{\mathrm{type}} + b_{\mathrm{AE}}^{(\ell)} + E_{\mathrm{med}\to\mathrm{AE}}^{(\ell)}[m_t]\}}{\sum_{j=1}^{L} \exp\{(U_{\mathrm{AE}}\mathbf{s}_t)_j + \beta_{\mathrm{trend}}^{(j)}\mathrm{trend}_t + \kappa_{c,j}^{\mathrm{type}} + b_{\mathrm{AE}}^{(j)} + E_{\mathrm{med}\to\mathrm{AE}}^{(j)}[m_t]\}}, \tag{13}$$

where $U_{\mathrm{AE}} \in \mathbb{R}^{L \times 4}$, $\beta_{\mathrm{trend}}^{(\ell)} \geq 0$ provides a gentle bias under worsening trends, $\kappa_{c,\ell}^{\mathrm{type}}$ is an arm-specific shift, $b_{\mathrm{AE}}^{(\ell)}$ is a type bias, and $E_{\mathrm{med}\to\mathrm{AE}}^{(\ell)}[m]$ captures med-type–specific effects.

**Laboratory measurement** The lab is a linear readout of the state plus an immediate medication offset and noise:

$$y_t = \mathbf{h}^\top \mathbf{s}_t + w_{m_t} + \epsilon_t^{\mathrm{lab}}, \qquad \epsilon_t^{\mathrm{lab}} \sim \mathcal{N}(0, \sigma_{\mathrm{lab}}^2), \quad \sigma_{\mathrm{lab}} = 0.10. \tag{14}$$

We set $w_k < 0$ for $k \geq 1$ so that medications immediately lower the lab.

**Within-visit causal order** We respect the following order each visit:

$$\boxed{y_{t-1} \ \longrightarrow\ m_t \ \longrightarrow\ (a_t,\, y_t) \ \longrightarrow\ \mathbf{s}_{t+1}}$$

i.e., the previous lab informs prescribing; the current medication affects both AE risk and the current lab; and medications impart a persistent effect on the next state via equation 9.

## A.3 DETAILS FOR BREAST CANCER TRIAL DATA

**Data processing.** The time-dependent visit sequences include treatments, medications, adverse events, and serious adverse events, all extracted from the raw clinical trial data. We define a target variable severe outcome, which represents the experience of any serious adverse event or death throughout the complete visit history. Following Das et al. (2023), we keep the top 100 frequent medications and top 50 frequent adverse events, while combining all remaining adverse events as one feature representing all rare adverse events. For static features, we impute missing values for numeric and categorical features by mean and mode, respectively. To ensure sufficient temporal context, we filter out patients with fewer than five recorded visits. Finally, we randomly assign 70% and 30% of the patients to training and test sets, respectively.

**Summary table.**

Table 4: Summary of the breast cancer trial dataset

| Item | Value |
| --- | --- |
| Total # of patients | 941 |
| Min visit length | 6 |
| Max visit length | 17 |
| Mean visit length | 10 |
| # of Treatments | 2 |
| # of Medications | 100 |
| # of Adverse events | 50 |
| # of Numerical features | 7 |
| # of Binary features | 2 |
| # of Categorical features | 7 |
| % of patients with severe outcome | 13.6% |

