# OpenReview forum: "Counterfactual Digital Twin: Generating What-If Trajectories with Uncertainty"
_ICLR.cc/2026/Conference — Submitted to ICLR 2026_

### Official Review · Reviewer_bkgq · 2025-10-31

**Soundness:** 3
**Presentation:** 3
**Contribution:** 3
**Rating:** 6
**Confidence:** 3

**Summary:**

This paper aims to help provide full rundowns of multiple alternative versions of input scenarios, supported by clearly traceable reasoning.

**Strengths:**

The authors have provided a detailed rundown of the mathematical processes they use to calculate these trajectories, and display a great understanding of the concepts they discuss.

**Weaknesses:**

There should ideally be a little more detail given in the beginning about why some algorithms were chosen over others.

**Questions:**

There should ideally be a little more detail given in the beginning about why some algorithms were chosen over others.
For example, why TFT over Informer (https://arxiv.org/abs/2012.07436), or pinball loss over a potentially more robust alternative (https://arxiv.org/pdf/2011.09588)?
The first part of this paper would strongly benefit from a diagram to provide a simple framework example of how this methodology can be applied to create multiple counterfactual trajectory paths, as parts of the current text read a bit like a walkthrough of a series of mathematical methods without callbacks to why these choices were made.
While the last third of the paper does provide an experimental study within a medical context, the first two thirds read as a dense walkthrough of various mathematical concepts rather than an overview of a connected methodology.
I also did not see a clear definition of uncertainty within the context of this paper, which is important when using a term that has multiple definitions.
Additionally, if KNNSampler consistently performs better than CounterTwin, why is CounterTwin necessary? In what ways does it provide a better alternative to KNNSampler?

Overall, I rate this paper a weak accept. I think the idea is novel and has the capacity to address a variety of different real-world issues, particularly in the medical field. However, it needs to be more specific about the problems previous approaches have, as well as specify how CounterTwin solves these problems.
As a note, there may potentially be a typo on page 3: should one of the T expressions have a different suffix (not i), since it’s representing an alternate treatment path?

---

### Official Review · Reviewer_pmym · 2025-11-01

**Soundness:** 2
**Presentation:** 2
**Contribution:** 2
**Rating:** 2
**Confidence:** 3

**Summary:**

The paper introduces CounterTwin, a "counterfactual digital twin" framework that generates personalized what-if trajectories for an individual under different treatment scenarios, along with estimates of uncertainty. Instead of simulating a single observed trajectory, CounterTwin aims at produce multiple treatment-conditioned paths for the same unit (for instance a considered patient) and quantify the confidence in those prediction.  The contribution consists in formalizing the setting of conterfactual traectories generation with a digital twin, defining CounterTwin that combines quantile-level sequence predictions of individual progression with SC-based donor distributions, and integrates them via a Kalman update to produce treatment replicas with posterior uncertainty. Countertwin performance is evaluated on both synthetic and real-world data.

**Strengths:**

- the main paper is overall well written ad easy to read.
- The considered topic of evaluating counterfactual trajectories over time is definitely relevant and the idea of combining this with modern architectures and UQ methods is nice
- the experimental section combines both synthetics and real world examples

**Weaknesses:**

- the related work sections (and consequently baselines choice) is missing a major literature area that focuses on the formalization of structural causal models within dynamic treatment regimens for counterfactual trajectories generation. See below in questions
- the paper does not mention any commonly made causal assumptions on the system, for instance consistency, sequential ignorability, and sequential overlap and then there are not theoretical results that its counterfactual estimates are identifiable from data. This is not necessary in all cases but I think in these settings then the claims need to be toned down and the experimental section more complete.

**Questions:**

- can the authors elaborate on the methodological difference between this approach and methods like [1,2,3,4] and specifically whether the  identifiability guarantees are similar?


[1] Melnychuk, Valentyn, Dennis Frauen, and Stefan Feuerriegel. "Causal transformer for estimating counterfactual outcomes." International conference on machine learning. PMLR, 2022.
[2] Bica, Ioana, et al. "Estimating counterfactual treatment outcomes over time through adversarially balanced representations." arXiv preprint arXiv:2002.04083 (2020).
[3] Li, Rui, et al. "G-net: a recurrent network approach to g-computation for counterfactual prediction under a dynamic treatment regime." Machine Learning for Health. PMLR, 2021.
[4] Schwarz, Thomas, Cecilia Casolo, and Niki Kilbertus. "Uncertainty-Aware Optimal Treatment Selection for Clinical Time Series." arXiv preprint arXiv:2410.08816 (2024).

---

### Official Review · Reviewer_Ap1p · 2025-11-02

**Soundness:** 1
**Presentation:** 2
**Contribution:** 2
**Rating:** 2
**Confidence:** 3

**Summary:**

The paper introduces CounterTwin, an approach that combines TTF (a transformer model for forecasting patient trajectories) and the synthetic control framework to generate uncertainty-aware counterfactuals. The approach is evaluated on both synthetic and real clinical data.

**Strengths:**

* The problem of generating counterfactuals in digital twins is interesting and relevant
* The idea of combining forecasting models, synthetic controls, and Kalman filter fusion seems novel

**Weaknesses:**

The paper has two main weaknesses: insufficient justification of some design choices and shortcomings in the experimental evaluation. Regarding design choices:
* One is the choice of the TFT model. TFT outputs quantiles, but you need mean and variance. Have you considered architectures that directly output the parameters of the Gaussian?
* Another is the choice of a Kalman filter for combining TFT and SC outputs. KFs are optimal only when the underlying system is linear and may perform poorly otherwise, with predicted covariance blow-ups. Why not consider a particle filter / rejection sampling-based approach, which doesn't require such assumptions?
* The choice of uncertainty sources in the SC model is sensible, but arbitrary. Do you have empirical evidence to support this particular design, as opposed to e.g, deterministic weights (i.e., baseline SC) or without $\Sigma_\text{mis}$?

Regarding the experimental evaluation:
* The analysis of counterfactuals (Table 2 and Figure 4) should also include the performance of your SC method, to understand if the fusion with the TTF data brings any advantage.
* Moreover, Figure 4 should include results with SynCtrl (2021)
* Since it's a central part of the approach, it's important to assess the quality of the Kalman-based fusion on the real data, where process linearity is not guaranteed. It'd be good to assess, e.g., how the predicted covariance evolves over time and how variance distributes across the different components/dimensions of the state.
* Section 6 is not useful; figures 5 and 6 don't contribute to validating the method and are not adequately explained.

**Questions:**

I'd appreciate it if you could address the questions and issues raised in the weaknesses section.

---

### Meta-Review · Area_Chair_19KD · 2026-01-07

**Summary:**

The paper introduces an approach based on a transformer model and synthetic control to generate uncertainty-aware counterfactuals, and validates the approach on synthetic and real clinical data. The reviewers brought a number of concerns regarding the design choices, assumptions and experimental evaluation, and a majority of reviewers was in favor of rejecting the paper. I encourage the authors to revise their paper in light of the reviews and resubmit to another venue.

**Reviewer Scores:**

I do not think the discussion would have significantly change their scores.

---

### Decision · Program_Chairs · 2026-01-26

Reject